# Associations between sounds and actions in early auditory cortex of nonhuman primates

**Ying Huang[1,2]\*, Peter Heil[2,3], Michael Brosch[1,2]**

[1]Special Lab Primate Neurobiology, Leibniz Institute for Neurobiology, Magdeburg, Germany; [2]Center for Behavioral Brain Sciences, Otto-von-Guericke-University, Magdeburg, Germany; [3]Department Systems Physiology of Learning, Leibniz Institute for Neurobiology, Magdeburg, Germany

**Abstract** An individual may need to take different actions to the same stimulus in different situations to achieve a given goal. The selection of the appropriate action hinges on the previously learned associations between stimuli, actions, and outcomes in the situations. Here, using a go/no-go paradigm and a symmetrical reward, we show that early auditory cortex of nonhuman primates represents such associations, in both the spiking activity and the local field potentials. Sound-evoked neuronal responses changed with sensorimotor associations shortly after sound onset, and the neuronal responses were largest when the sound signaled that a no-go response was required in a trial to obtain a reward. Our findings suggest that association processes take place in the auditory system and do not necessarily rely on association cortex. Thus, auditory cortex may contribute to a rapid selection of the appropriate motor responses to sounds during goal-directed behavior.

DOI: https://doi.org/10.7554/eLife.43281.001

## Introduction

Imagine you stand at a crosswalk and hear the honking of a car. If you are at the edge of the crosswalk, you should stop walking to avoid collision, but if you are already in the middle of the crosswalk, you should speed up walking for the same reason. The appropriate action to a given sensory stimulus can therefore depend on the situation, and it may be necessary to select it rapidly. The selection of the action hinges on the previously learned associations between stimuli, actions, and outcomes in the situations. Previous studies have shown that neurons in frontal and prefrontal cortices respond differently to a given stimulus when it signals different actions (*Asaad et al., 1998*; *Komatsu, 1982*; *Sakagami and Tsutsui, 1999*; *Sakagami et al., 2001*; *Watanabe, 1986*; *Yamatani et al., 1990*). However, very little is known whether similar representations of associations between stimuli and actions (*sensorimotor associations*) are present also in sensory cortices. If so, this might accelerate the selection of the appropriate action because such representations might be directly conveyed to motor cortex via anatomical projections from sensory cortex to motor cortex (*Matyas et al., 2010*; *Scheich et al., 2007*), and not only indirectly via cortical association areas.

To our knowledge, only few studies have investigated whether learned sensorimotor associations are represented in sensory cortex and specifically in auditory cortex, which is known to be more than a mere stimulus processor (*Scheich et al., 2011*; *Weinberger, 2011*). Several studies reported that neurons in early auditory cortex responded differently to a given sound depending on the action the animal had to execute in response to the sound (*David et al., 2012*; *Jaramillo et al., 2014*; *Vaadia et al., 1982*). However, the observed differences in the neuronal responses do not necessarily reflect differences in sensorimotor associations but perhaps differences in confounding factors

\*For correspondence:
Ying.Huang@lin-magdeburg.de

**Competing interests:** The authors declare that no competing interests exist.

known to affect neuronal activity in auditory cortex. These factors include the events used to cue the animal about the association between sound and action (*Brosch et al., 2005*; *Brosch et al., 2015*; *Rodgers and DeWeese, 2014*; *Vaadia et al., 1982*), the acoustic context in which the sound was embedded (*Brosch and Scheich, 2008*; *Brosch and Schreiner, 1997*; *Brosch et al., 1998*; *Brosch et al., 1999*; *Jaramillo et al., 2014*; *Nelken and Ulanovsky, 2007*; *Ulanovsky et al., 2003*), and the valence of the sound, that is the associated type of reinforcement (*Blake et al., 2006*; *David et al., 2012*; *Rutkowski and Weinberger, 2005*). It is therefore still an open question whether sensory cortex represents sensorimotor associations.

Here, we addressed this question by recording neuronal activity from early auditory cortex of nonhuman primates while they performed tasks which required them to execute different actions to the same sound to receive a reward. We recorded not only action potentials but also local field potentials. The latter allowed us to also investigate whether sensorimotor associations are represented in the summed graded and action potentials of neuronal populations in auditory cortex.

## Results

### Experimental rationale and overview

Two macaque monkeys were each trained to perform two tasks requiring different sensorimotor associations. In both tasks, a trial started with the illumination of a light emitting diode (LED) where upon the monkeys had to grasp and hold a touch bar for several seconds (*Figure 1A*). This triggered the presentation of a sequence of two 200 ms tones, S1 and S2, separated by an 800 ms delay. Each tone could have a frequency of either 3 kHz (*Figure 1B*; orange boxes) or 1 kHz (blue boxes), resulting in the four sequences 3–3, 3–1, 1–3, and 1–1. Depending on the sequence and the task, the monkeys were required to either release the bar within a specific time interval after S2 (a *go response*) or to continue holding the bar and thus not to release the bar within this interval (a *no-go response*) to obtain a reward (a small amount of water). The monkeys received the reward for correct go responses (*hits*) and correct no-go responses (*correct rejections*) and no reward for incorrect responses (*misses* and *false alarms*).

In Task 1, the go response was required for the sequence 3–3 and the no-go response for the other three sequences (3–1, 1–3, and 1–1; *Figure 1B*). The sequence presented in a given trial was chosen randomly such that at the start of the trial the monkeys could not know which motor response would be required to obtain the reward. Identification of the frequency of S1 was a necessary first step toward resolving this uncertainty. However, when S1 was 3 kHz, identification of the frequency of S1 was not sufficient: even an ideal observer would remain uncertain (*S1-uncertain*; *Figure 1C*, top). The sufficient information was provided by S2, with the go response required when S2 was also 3 kHz (*S2-go*) and the no-go response when it was 1 kHz (*S2-no-go*). In contrast, when S1 was 1 kHz, identification of the frequency of S1 was both necessary and sufficient for resolving the uncertainty and realizing that the no-go response was required (*S1-no-go*; *Figure 1C*, bottom). In this case, identification of the frequency of S2 was not necessary because the no-go response was required irrespective of the frequency of S2 (*S2-nil*). Therefore, the two tones used as S1 differed with respect to whether they could resolve the uncertainty regarding the required motor response and thus could signal the motor response or not (S1-no-go vs. S1-uncertain). In the sequences where S1 was 3 kHz (*Figure 1C*, top), the two tones used as S2 differed with respect to whether they signaled the go response or the no-go response (S2-go vs. S2-no-go). In the sequences where S1 was 1 kHz (*Figure 1C*, bottom), neither of the two S2-nils was needed to signal the motor response (the no-go response).

We note that the behavioral meaning of a given tone can be formulated in two different, albeit equivalent, ways. For example, S2-go means reward for the go-response and equivalently no reward for the no-go response. Similarly, S2-no-go means reward for the no-go-response and equivalently no reward for the go response. S2-go and S2-no-go therefore differed with respect to the motor response required to obtain the reward (e.g., go: reward vs. no-go: reward) and with respect to the reward value of a given motor response (e.g., go: reward vs. go: no reward). Our experimental design cannot distinguish between these two equivalent formulations of a stimulus-action-outcome association. For simplicity, we use the term sensorimotor association for both. Moreover, sequences where S1 was 1 kHz had lower memory demands during the delay than sequences where S1 was 3

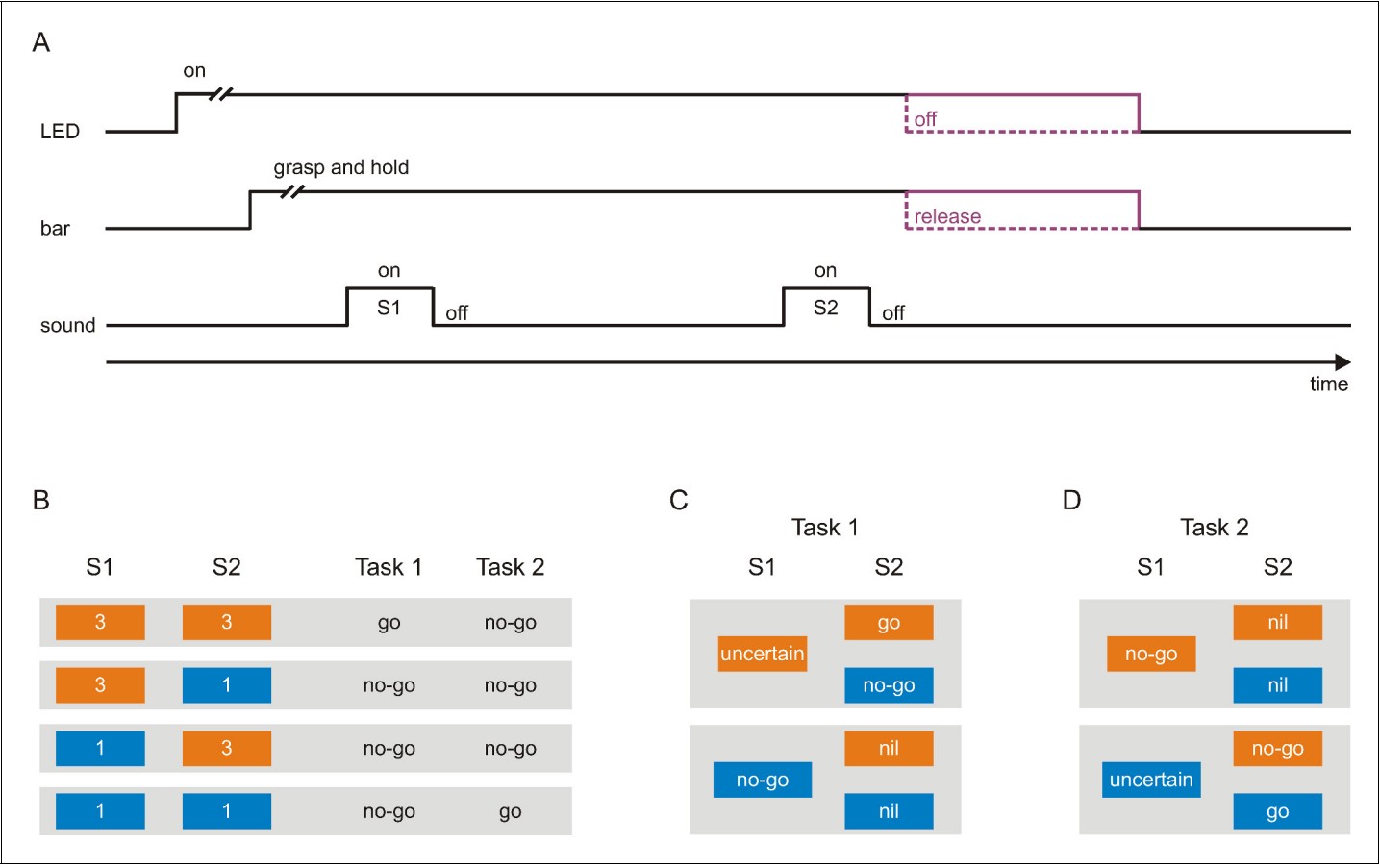

**Figure 1.** Scheme of the tasks used to investigate neuronal representations of sensorimotor associations in early auditory cortex. (**A**) A trial started with the illumination of a light-emitting diode (LED), after which the monkeys had to grasp and to hold a touch bar for some time. This triggered the presentation of a sequence of two tones, S1 and S2, separated by a delay. The monkeys were required to either release the bar within a specific interval after S2 (*go response*) or to continue holding the bar and thus not to release the bar within this interval (*no-go response*). Bar release caused the LED to turn off. The monkeys were rewarded with a small amount of water for correct go responses and correct no-go responses. (**B**) Each tone could have a frequency of either 3 kHz (orange boxes) or 1 kHz (blue boxes), resulting in the four possible sequences 3–3, 3–1, 1–3, and 1–1. In Task 1, the go response was required for the sequence 3–3 and the no-go response for the other sequences. In Task 2, the go response was required for the sequence 1–1 and the no-go response for the other sequences. The monkeys were cued which task to perform in a trial, with a green LED located to the monkey's right for Task 1 and with a red LED located to the monkey's left for Task 2. (**C–D**) Sensorimotor associations of S1s and S2s in Task 1 (**C**) and Task 2 (**D**) from the perspective of an ideal observer. At the start of a given trial, the observer could not know which motor response would be required to obtain a reward, and identification of the frequency of S1 was a necessary first step towards resolving this uncertainty. However, in Task 1, when S1 was 3 kHz, identification of S1 was not sufficient and S1 therefore could not signal the required motor response (*S1-uncertain*). The sufficient information was provided by S2, with the go response signaled by S2 when it was 3 kHz (*S2-go*) and the no-go response when it was 1 kHz (*S2-no-go*). When S1 was 1 kHz, identification of S1 was both necessary and sufficient for resolving the uncertainty and therefore S1 could signal the required motor response (*S1-no-go*). In this case, identification of S2 was not necessary for resolving the uncertainty and S2 was therefore not needed to signal the motor response (*S2-nil*). In Task 2, in trials where S1 was 1 kHz, S1 could not signal the required motor response (*S1-uncertain*) and S2 signaled the go response when it was also 1 kHz (*S2-go*) and the no-go response when it was 3 kHz (*S2-no-go*). In trials where S1 was 3 kHz, S1 could signal the required motor response (*S1-no-go*) and S2 was not needed to signal the motor response (*S2-nil*).

DOI: https://doi.org/10.7554/eLife.43281.002

The following figure supplement is available for figure 1:

**Figure supplement 1.** Both monkeys discriminated between the go sequence and each of the no-go sequences.

DOI: https://doi.org/10.7554/eLife.43281.003

kHz, because only in the latter sequences the two possible sensorimotor associations of S2 had to be memorized to solve the task. These differences in memory demands were used in a previous study to demonstrate that there is neuronal activity in auditory cortex during the delay related to working memory (*Huang et al., 2016a*). Finally, sequences where S1 was 1 kHz and where it was 3

kHz might have also differed in the level of attention after S1, but not during S1, because the monkeys could solve the task solely by identifying S1 in the former trials while they remained uncertain after S1 and needed to identify S2 to solve the task in the latter trials.

Although Task 1 required different motor responses to different tones (i.e., different sensorimotor associations), the required motor responses co-varied with the frequency of the tones and with the acoustic context in which the tones were embedded. We therefore designed Task 2 in which the same set of tone sequences was used but the required motor responses for the sequences 3–3 and 1–1 were reversed relative to those in Task 1, that is the go response was required for the sequence 1–1 and the no-go response for the remaining sequences (*Figure 1B*). Therefore, the sensorimotor associations required in Task 2 (*Figure 1D*) differed from those in Task 1 (*Figure 1C*).

This experimental design should enable testing whether sensorimotor associations are represented in auditory cortex by comparing neuronal responses to tones of the same frequency and in corresponding positions in the two tasks (compare *Figure 1C and 1D*). In the following, we first compare neuronal responses to S1-no-go and S1-uncertain. We then compare neuronal responses to S2-go, S2-no-go and S2-nil. We limit our analyses to short periods after tone onset because neuronal activity in auditory cortex can be related to the actual execution of motor behavior (*Brosch et al., 2005*; *Eliades and Wang, 2008*). We also characterize the neuronal activity related to the execution of the go response and assess its potential contribution to differences in the neuronal responses to the tones between the tasks.

We cued the monkeys regarding which task to perform in a trial by illumination of different LEDs: a green LED to the monkey's right signaled Task 1 and a red LED to its left signaled Task 2. LED illumination also served to mark the start of a trial. Because visual stimuli, eye position, and the mental state related to them might result in phasic or tonic changes of neuronal activity in auditory cortex (*Bizley and King, 2008*; *Brosch et al., 2005*; *Brosch et al., 2015*; *Fu et al., 2004*; *Rodgers and DeWeese, 2014*; *Werner-Reiss et al., 2003*), we assessed potential contributions of the visual cue to differences in the neuronal responses between the tasks by analyzing the neuronal activity during periods directly before S1 and before S2.

Within each experimental session, the monkeys conducted the two tasks in separate, alternating blocks of trials. We used a block rather than a random design to increase the monkeys' performance. A block consisted of ~140 trials and lasted ~25 min. In each block, trials requiring the go response or the no-go response were presented in pseudorandom order, with the go response required in ~60% of the trials (range 50–77%) and the no-go response in ~40% of the trials (range 23–50%). Ideally, each sequence would have been presented with the same probability. However, this was not feasible because then the monkeys likely would have adopted a strategy of executing the no-go response in every trial, which would have yielded a reward in 75% of the trials. We therefore had to increase the percentage of trials requiring the go response. Consequently, the median probability for a tone to have a frequency of 3 kHz in Task 1 was about three times (range 2–6) that in Task 2, and the probability for a tone to have a frequency of 1 kHz in Task 1 was about one third that in Task 2. These differences in the probabilities of the 3 kHz and the 1 kHz tones within tasks and across tasks contributed to differences in the *acoustic context* and could result in different degrees of stimulus-specific adaptation of the neuronal responses (*Nelken and Ulanovsky, 2007*; *Ulanovsky et al., 2003*). Differences in stimulus-specific adaptation could arise irrespective of whether the monkeys performed the tasks or not. To estimate the task-independent effects of stimulus-specific adaptation, we recorded neuronal responses to the same set of sequences while the monkeys did not perform the tasks and did not receive the reward. In such *passive conditions*, the sequences were presented with an interval drawn randomly from a uniform distribution between 3.5 and 4.5 s and were organized in separate blocks. In half of these blocks, the four sequences were presented with the same probabilities as in Task 1 and in the other blocks with the same probabilities as in Task 2. In addition, we compared the neuronal responses from task sessions with small and large differences in the probabilities of the tones.

Microelectrode recordings of neuronal activity in auditory cortex started once the monkeys had responded correctly to each of the four sequences in ≥65% of the trials in both tasks during ten consecutive sessions. Single-unit and multiunit spiking activity as well as LFPs were recorded from the core fields of the right auditory cortex of both monkeys. This report is based on the data recorded during 243 experimental sessions. During 154 of these sessions, the monkeys performed the two tasks (108 and 46 sessions from monkey C and L, respectively). During 97 of these sessions, the

monkeys were also passively exposed to the tone sequences after they had performed the tasks. During another 89 sessions, the monkeys were only passively exposed to the tone sequences. During nearly all the 154 task sessions, the monkeys discriminated between the go and each of three no-go sequences in both tasks. This is shown using signal-detection theory. The d-prime values were calculated using the hit rate for the go sequences and the false-alarm rate for each of the no-go sequences for each session and task (*Figure 1—figure supplement 1*).

In the following sections, we focus on the results obtained from the 3 kHz tones. The corresponding results obtained from the 1 kHz tones were very similar, and they are shown in the supplement (*Figure 2—figure supplements 1–5*, *Figure 3—figure supplements 1–2*, and *Figure 5—figure supplement 1*).

## Representation of sensorimotor associations in auditory cortex revealed by comparing spike responses to S1

Many neurons in early auditory cortex responded differently to an S1 of a given frequency depending on whether S1 could signal the required motor response (S1-no-go) or not (S1-uncertain). *Figure 2A and 2B* show the spike rates of a representative multiunit averaged across all correctly performed trials. For this example, the spike rate during the presentation of an S1 of 3 kHz (orange traces) was higher in Task 2 (*Figure 2Bi*) than in Task 1 (*Figure 2Ai*), that is the spike rate was higher to S1-no-go than to S1-uncertain. To quantify the difference in the spike responses, we computed the ratio of the mean spike rate during S1-no-go to the mean spike rate during S1-uncertain (*during-S1 spike ratio*). A ratio >1 therefore represents a higher spike rate during S1-no-go than during S1-uncertain. The mean spike rate was computed over a 250 ms period starting with S1 onset and ending 50 ms after S1 offset to include offset responses. For this multiunit and for the 3 kHz tone, the during-S1 spike ratio was 1.40, which was significantly >1 (p<0.05, permutation test). This was not the case for the spike ratio computed from the mean spike rates during the corresponding 250 ms periods directly before S1 (*before-S1 spike ratio*).

*Figure 2Ci* shows the during-S1 spike ratios for all 298 multiunits for trials where S1 was 3 kHz and where the two monkeys correctly performed the two tasks (green curve). These ratios ranged from 0.38 to 2.78, with a median of 1.09 which was significantly >1 (p<0.05, Wilcoxon signed rank test). Ratios significantly different from 1 (that is >1 or <1; green dots) were obtained for 124 of the 298 multiunits (41.6%; 95 of the 226 multiunits [42.0%] for monkey C and 29 of the 72 multiunits [40.3%] for monkey L), with the majority of these ratios being >1 (81.5%). The occurrence of significant during-S1 spike ratios was related to the distance between the best frequencies of the multiunits and the frequency of the tested tone (for determination of best frequency, see Materials and methods). Significant during-S1 spike ratios occurred more frequently in the 35 multiunits with a best frequency within a range of ±0.5 octaves around 3 kHz than in the other 263 multiunits (57.1% vs. 39.5%; p<0.05, chi-square test). Significant during-S1 spike ratios were also found in the single units isolated from the multiunit recordings, although with lower incidence (20.8% of 144 single units for trials where S1 was 3 kHz; green dots in *Figure 2—figure supplement 2A*, top).

The following analyses suggest that the observed differences in the spike responses to the same S1 between Task 1 and Task 2 were at least partially related to whether S1 could signal the required motor response or not, because they could not fully be accounted for by the differences in the visual cue or in the acoustic context. To assess potential differences in tonic neuronal activity related to the visual cue between the two tasks and to estimate their contributions to the differences in the spike responses to S1, we analyzed the before-S1 spike ratios. This analysis revealed that the spike rates before S1 could indeed differ between the two tasks (black curve in *Figure 2Ci* for trials where S1 was 3 kHz). However, the median of the distribution was 0.97, which was not significantly different from 1 (p>0.05, Wilcoxon signed rank test). Also, significant before-S1 spike ratios (black dots) were observed less frequently than significant during-S1 spike ratios (88 of 298, 29.5% versus 124 of 298, 41.6%; p<0.05, chi-square test, one-tailed), and significant before-S1 spike ratios >1 were not over-represented (33 of 88, 37.5%). For the multiunits having significant during-S1 spike ratios, the during-S1 spike ratios were generally larger than the corresponding before-S1 spike ratios (*Figure 2Cii*), and for only a minority of these multiunits were the before-S1 spike ratios also significant (47 of 124, 37.9%; filled dots) while for the majority they were not (77 of 124, 62.1%; open dots). These results show that differences in spike rate related to the visual cue between the two tasks could account for significant during-S1 spike ratios in only maximally 37.9% of the cases. These results also indicate

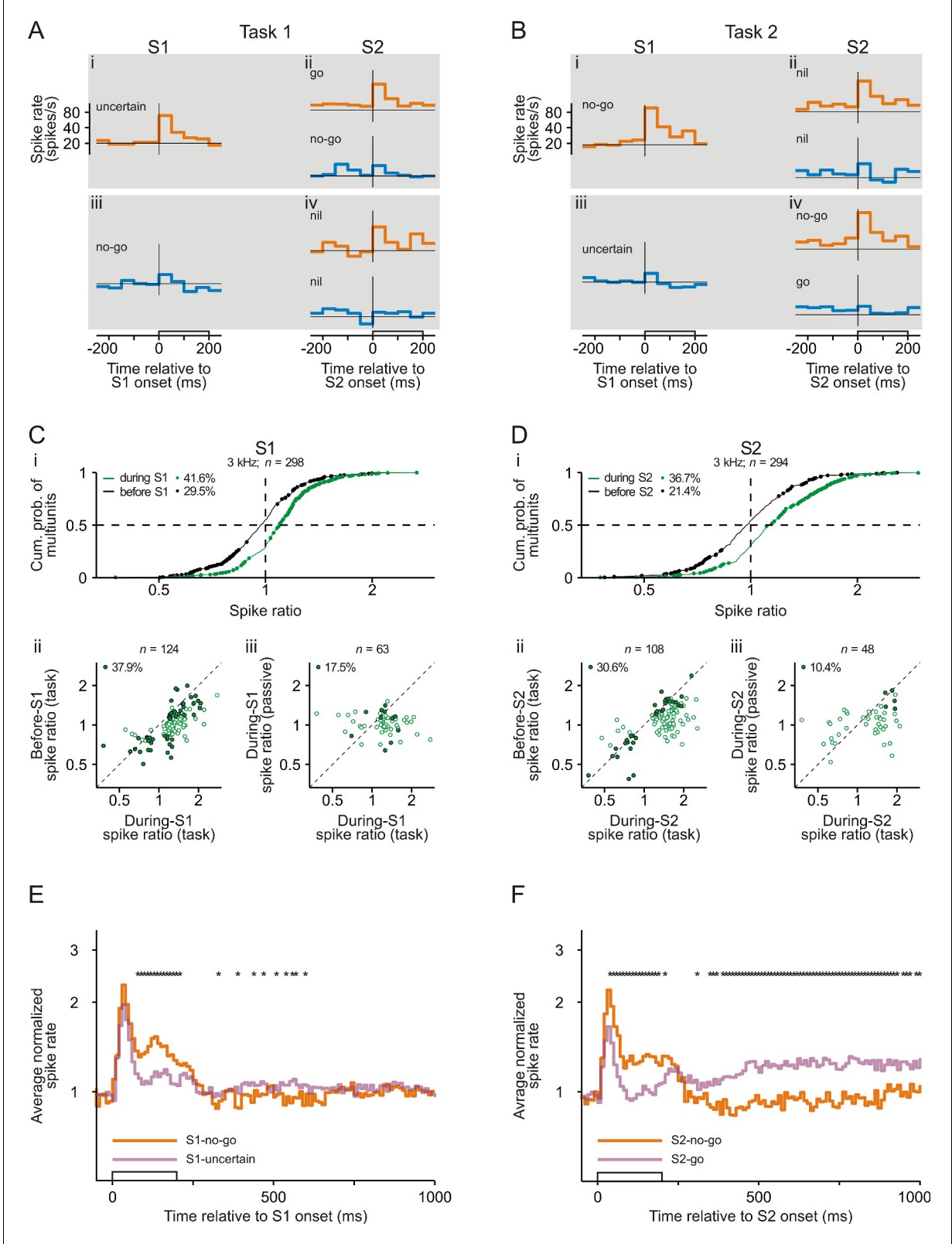

**Figure 2.** Representation of sensorimotor associations in the spiking activity in early auditory cortex. (**A–B**) Spike rates of an example multiunit in auditory cortex to S1 and S2 in Task 1 (**A**) and Task 2 (**B**). Orange and blue traces represent spike rates to tones of 3 kHz and of 1 kHz, respectively. Each trace shows the spike rate averaged across trials from 250 ms before tone onset to 50 ms after tone offset. The open bars on the abscissae mark the timing and the 200 ms duration of the tones. The horizontal lines represent a spike rate of 20 spikes/s. Note that the ordinates have logarithmic

*Figure 2 continued on next page*

*Figure 2 continued*

scaling. The sensorimotor associations of S1s and S2s are also provided. (**C**) Larger spike responses to S1-no-go than to S1-uncertain when the frequency of the tones was 3 kHz. (i) Cumulative distribution of during-S1 (green) and before-S1 (black) spike ratios for all 298 multiunits recorded from the two monkeys. During-S1 spike ratios were the ratios of the mean spike rates during S1-no-go in Task 2 to the mean spike rates during S1-uncertain in Task 1. Before-S1 spike ratios were the ratios computed from the mean spike rates during the corresponding periods directly before the S1s. Ratios significantly different from 1 are marked by dots, and the proportions of multiunits with such ratios are provided. The horizontal dashed line marks the cumulative probability of 0.5 and the vertical dashed line the ratio of 1. (ii) Larger during-S1 spike ratios than before-S1 spike ratios. The analysis was limited to the 124 multiunits having significant during-S1 spike ratios (green dots in panel i). The open dots mark multiunits with significant during-S1 ratios only and the filled dots mark multiunits with significant during-S1 and before-S1 ratios. The proportion of multiunits with significant during-S1 and before-S1 ratios is provided. The diagonal marks cases where the two ratios were equal. Note that the axes have logarithmic scaling. (iii) Larger during-S1 spike ratios during task performance than in the passive condition. This comparison was limited to the 63 multiunits that exhibited significant during-S1 spike ratios during task performance and that were tested also in the passive condition. The open dots mark multiunits with significant ratios during task performance only and the filled dots mark those with significant ratios both during task performance and in the passive condition. Other conventions are equivalent to those of panel Cii. (**D**) Larger spike responses to S2-no-go than to S2-go when the frequency of the tones was 3 kHz. The organization of the panel and other conventions are equivalent to those of panel C. During-S2 spike ratios were the ratios of the mean spike rates during S2-no-go in Task 2 to the mean spike rates during S2-go in Task 1. Before-S2 spike ratios were the ratios computed from the mean spike rates during the corresponding periods directly before the S2s. (**E**) The average normalized spike rates to S1-no-go (orange trace) and S1-uncertain (purple trace) of the 124 multiunits with significant during-S1 spike ratios for the 3 kHz tone, shown from 50 ms before S1 onset to 800 ms after S1 offset. They were obtained by geometrically averaging the spike rates after normalizing each rate to baseline (the mean spike rate during the 250 ms period directly before S1). Note the logarithmic scaling of the ordinate. The stars above the traces mark a significant difference (p<0.05) between the two traces. (**F**) The average normalized spike rates to S2-no-go (orange trace) and to S2-go (purple trace) of the 108 multiunits with significant during-S2 spike ratios for the 3 kHz tone. Other conventions are equivalent to those of panel E.

DOI: https://doi.org/10.7554/eLife.43281.004

The following figure supplements are available for figure 2:

**Figure supplement 1.** Representation of sensorimotor associations in early auditory cortex revealed by comparing the spike responses to the tones with a frequency of 1 kHz.

DOI: https://doi.org/10.7554/eLife.43281.005

**Figure supplement 2.** Distributions of the spike ratios of single units in early auditory cortex.

DOI: https://doi.org/10.7554/eLife.43281.006

**Figure supplement 3.** No consistent effects of tone probability on the differences in the neuronal responses between Task 1 and Task 2.

DOI: https://doi.org/10.7554/eLife.43281.007

**Figure supplement 4.** Spiking activity related to bar release in early auditory cortex.

DOI: https://doi.org/10.7554/eLife.43281.008

**Figure supplement 5.** Comparison of spike rates to S2 when S2 was needed to signal a motor response or not.

DOI: https://doi.org/10.7554/eLife.43281.009

that the significant during-S1 spike ratios could not simply be explained by spontaneous fluctuations in spike rate between the tasks that were performed in separate blocks.

To estimate potential contributions of the difference in the acoustic context to the differences in the spike responses to S1 between the two tasks, we analyzed the during-S1 spike ratios from the two corresponding passive conditions where the probabilities of the 3 kHz and the 1 kHz tones were the same as in the tasks. For the 3 kHz tone, this analysis was performed on 473 multiunits. Significant during-S1 spike ratios occurred less frequently in the passive condition than during task performance (19.2% vs. 41.6%; p<0.05, chi-square test, one-tailed). Sixty-three of the 124 multiunits having significant during-S1 spike ratios during task performance were also tested in the passive condition and generally had smaller during-S1 spike ratios in the passive condition (*Figure 2Ciii*). Only a minority had significant during-S1 spike ratios also in the passive condition (11 of 63, 17.5%; filled dots) while the majority did not (52 of 63, 82.5%; open dots). This indicates that 17.5% of the significant during-S1 spike ratios during task performance might be explained by the difference in the acoustic context between the two tasks (i.e., by task-independent stimulus-specific adaptation).

Further support for the suggestion that the significant during-S1 spike ratios during task performance were not solely due to the difference in the acoustic context was obtained by comparing during-S1 spike ratios from experimental sessions having different probabilities of the 3 kHz and the 1 kHz tones. The median probability of the 3 kHz tone across all sessions was 0.73 in Task 1 and 0.27 in Task 2. Task-independent stimulus-specific adaptation might therefore cause weaker spike responses to the 3 kHz tone in Task 1, resulting in during-S1 spike ratios predominantly >1. If such

stimulus-specific adaptation were the only mechanism, then during-S1 spike ratios should be larger when computed from the subset of the sessions where the ratio of the probabilities of the 3 kHz tone in Task 1 and Task 2 was greater than the median ratio of 2.70 (0.73/0.27) than when computed from the subset of the sessions where that ratio was smaller than the median ratio. This however was not the case. The proportion of multiunits with significant during-S1 spike ratios >1 did not differ between these two subsets of sessions in the direction predicted from stimulus-specific adaptation (*Figure 2—figure supplement 3A*, top left).

These analyses enable a conservative estimate of the percentage of multiunits in early auditory cortex that represent sensorimotor associations. Because 37.9% of the significant during-S1 spike ratios during task performance might be explained by the difference in the visual cue and 17.5% by the difference in the acoustic context, the remainder (55 of 124; 44.4%) provides the minimum percentage of significant during-S1 spike ratios that were due to the difference with respect to whether S1 could signal the required motor response or not. Overall, therefore, at least 18.5% of the multiunits (55/298) in early auditory cortex represent sensorimotor associations of S1.

To examine when the spike rates to S1-no-go and S1-uncertain differed, we geometrically averaged the spike rates, computed over consecutive 10 ms bins, across multiunits with significant during-S1 spike ratios during task performance. The averaging was conducted after normalizing each rate to the mean spike rate during the 250 ms period directly before S1. The normalization corrected for differences in the spiking activity before S1 between the two tasks. The average normalized spike rates to S1-no-go and S1-uncertain started to differ ~70 ms after tone onset, persisted throughout the tone, and disappeared shortly after tone offset (*Figure 2E*, compare orange and purple traces). Differences in the average normalized spike rates reappeared after S1 offset but were of opposite sign and small, suggesting that they might be related to other aspects of the tasks. For example, the differences during the delay after S1 could be related to differences in working memory load (*Huang et al., 2016a*).

## Representation of sensorimotor associations in auditory cortex revealed by comparing spike responses to S2

Analysis of the spike responses to S2 provides further support for the idea that neurons in early auditory cortex represent sensorimotor associations because it shows that the spike responses depended on whether S2 signaled the go response or the no-go response. For example, for the 3 kHz tone, the multiunit in *Figure 2A and 2B* exhibited a higher spike rate to S2-no-go in Task 2 (*Figure 2Biv*, orange trace) than to S2-go in Task 1 (*Figure 2Aii*, orange trace). To quantify this difference in the spike responses, we computed the ratio of the mean spike rate during S2-no-go to the mean spike rate during S2-go (*during-S2 spike ratio*). A ratio >1 therefore represents a higher spike rate during S2-no-go than during S2-go. The mean spike rates were computed over a 250 ms period starting with S2 onset and ending 50 ms after S2 offset. The permutation test revealed that the during-S2 spike ratio for this multiunit was 1.33, which was significantly >1 (p<0.05). This was not the case for the *before-S2 spike ratio* computed from the mean spike rates during the corresponding 250 ms periods directly before S2.

During-S2 spike ratios for the 3 kHz tone (*Figure 2Di*, green curve) ranged from 0.38 to 3.11, with a median of 1.13 which was significantly >1 (p<0.05, Wilcoxon signed rank test). Significant during-S2 spike ratios (green dots) were obtained for 108 of 294 multiunits (36.7%). The majority of these significant ratios were >1 (80.6% of 108). Significant during-S2 spike ratios were also obtained for isolated single units, albeit with lower probability (11.1%; *Figure 2—figure supplement 2B*, top).

Using similar reasoning as for S1, the following observations indicate that some of the observed significant during-S2 spike ratios can only be explained by the difference in the motor response signaled by S2 but not by the difference in the visual cue or by the difference in the acoustic context. The latter included also the immediate stimulus history, because in S2-go trials, S1 had the same frequency as S2, but in S2-no-go trials, S1 and S2 had different frequencies. This difference in the immediate stimulus history could result in different degrees of forward suppression of the spike responses to S2 in the two tasks. To estimate the potential contributions of the difference in the visual cue to the difference in the spike responses between the two tasks, we analyzed the before-S2 spike ratios. Before-S2 spike ratios (*Figure 2Di*, black curve) ranged from 0.39 to 2.59, with a median of 0.98 which was not significantly different from 1 (p>0.05, Wilcoxon signed rank test). Significant before-S2 spike ratios (black dots) were observed less frequently than significant during-S2

spike ratios (21.4% vs. 36.7%; p<0.05, chi-square test, one-tailed; compare black and green dots in *Figure 2Di*), and significant before-S2 spike ratios >1 were not overrepresented (42.9%; 27 of the 63 significant ratios). For the 108 multiunits having significant during-S2 spike ratios, these ratios were generally larger than the corresponding before-S2 spike ratios (*Figure 2Dii*), and for only a minority of these multiunits were the before-S2 spike ratios also significant (33 of 108, 30.6%; filled dots). These results indicate that the difference in the visual cue between the two tasks could account for significant during-S2 spike ratios in maximally 30.6% of the cases.

To estimate the potential contributions of the difference in the acoustic context to the difference in the spike responses to S2 between the two tasks, we analyzed the during-S2 spike ratios from the corresponding two passive conditions. For the 3 kHz tone, significant during-S2 spike ratios occurred less frequently in the passive condition than during task performance (13.1% vs. 36.7%; p<0.05, chi-square test, one-tailed). Forty-eight of the 108 multiunits having significant during-S2 spike ratios during task performance were also tested in the passive condition. These multiunits exhibited generally smaller during-S2 spike ratios in the passive condition than during task performance (*Figure 2Diii*), and only a minority of them exhibited significant during-S2 spike ratios also in the passive condition (5 of 48, 10.4%; filled dots). This indicates that only 10.4% of the significant during-S2 spike ratios during task performance could be explained by the difference in the acoustic context between the two tasks (i.e., by task-independent stimulus-specific adaptation or forward suppression). We also computed during-S2 spike ratios from the two subsets of experimental sessions where the ratio of the probabilities of the 3 kHz tone in Task 1 and Task 2 was greater than the median ratio and where it was smaller than the median ratio. The proportion of multiunits with significant during-S2 spike ratios >1 hardly differed between the two subsets of sessions (*Figure 2—figure supplement 3A*, top right). These findings support the idea that the significant during-S2 spike ratios during task performance were not solely caused by the difference in the acoustic context between the two tasks.

We conclude that, during task performance, the significant during-S2 spike ratios from at least 64 of the 108 multiunits (59%) were due to the difference in the sensorimotor association of S2 (no-go versus go). Therefore, ~20% of the multiunits recorded in auditory cortex (64/294) represented sensorimotor associations of S2.

Similar to S1, the normalized spike rates to S2-go and S2-no-go, geometrically averaged across multiunits with significant during-S2 spike ratios, started to differ shortly after tone onset, persisted throughout the tone, and disappeared shortly after tone offset (*Figure 2F*, compare purple and orange traces). Differences in the average normalized spike rates reappeared after S2 offset but were of opposite sign. The differences in the spike rates after S2 offset were partly related to the fact that the monkeys released the bar after S2-go but held on to the bar after S2-no-go. *Figure 2—figure supplement 4A* shows the average normalized spike rates with reference to the time of bar release executed after S2-go, revealing that spike rates in auditory cortex increased and peaked shortly (~200 ms) before the bar release. The increase started more than 250 ms before bar release and therefore may have affected the differences in spike rates during the 250 ms period after tone onset between S2-go and S2-no-go, but would have decreased the chance of observing neuronal activity related to sensorimotor associations in auditory cortex.

We obtained additional support for the conclusion that auditory cortex represents sensorimotor associations by comparing the spike response to S2-nil in one task with the spike responses to S2-go and S2-no-go in the other task. Analyses similar to those described above revealed that multiunits generally responded more strongly to S2-no-go than to S2-nil, with a median spike ratio of 1.10 which was significantly >1 (p<0.05, Wilcoxon signed rank test; orange curve in *Figure 2—figure supplement 5A*; for an example multiunit, compare the orange traces in *Figure 2Biv* and *Figure 2Aiv*). However, there was no differences in the spike responses to S2-go and S2-nil: the median spike ratio was 1.00, not significantly different from 1 (p>0.05, Wilcoxon signed rank test; purple curve in *Figure 2—figure supplement 5A*; for an example multiunit, compare the orange traces in *Figure 2Aii* and *Figure 2Bii*). This makes an explanation in terms of differences in the level of attention between S2-no-go and S2-nil unlikely because S2-go and S2-no-go should require a similar level of attention. It has been shown that neuronal activity in auditory cortex varies with the level of attention for other behavioral tasks (e.g., *Fritz et al., 2003*).

## Spike responses were strongest to tones that signaled the no-go response

In previous sections, we found that the spike response to a given tone in a given position was stronger when the tone signaled that the no-go response was required in a trial relative to when it did not do this, that is the spike response was stronger to S1-no-go than to S1-uncertain, and stronger to S2-no-go than to S2-go and to S2-nil. Here, we asked whether the spike responses to the two no-go tones were also stronger than the responses to the other tones, irrespective of their positions in the sequences. The comparisons were limited to the subpopulation of multiunits that had either significant during-S1 spike ratios or significant during-S2 spike ratios or both. Because we were interested in the relative strength of the spike responses in different conditions and because spiking activity in auditory cortex could change between conditions before tone presentation, we compared the normalized spike responses, that is the mean spike rate during the 250 ms period after tone onset normalized to the mean spike rate during the 250 ms period directly before the tone.

*Figure 3A* shows the normalized spike responses to the 3 kHz tone in each of the six conditions for 151 multiunits. The normalized spike responses to S2-no-go were significantly stronger than those to S1-uncertain, S2-go and the two S2-nils and the normalized spike responses to S1-no-go were significantly stronger than those to S1-uncertain, S2-go, and the two S2-nils (p<0.05/4, each corrected for multiple pairwise comparisons, Wilcoxon signed rank test; compare orange and purple

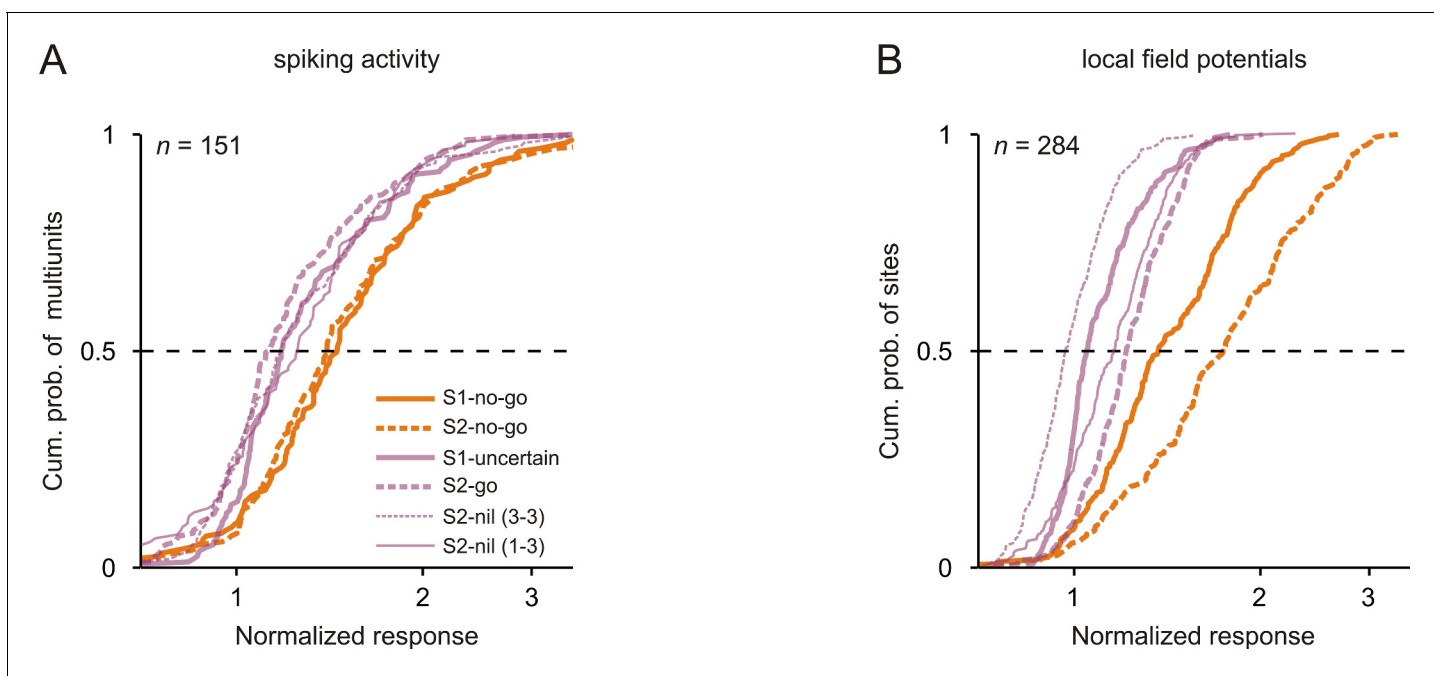

**Figure 3.** Neuronal responses were strongest to tones that signaled the no-go response. Cumulative distributions of the normalized neuronal responses to the 3 kHz tone in the six conditions (different sensorimotor associations and positions) for spiking activity (**A**) and local field potentials (**B**). For each tone, the normalized neuronal response was calculated as the mean activity during the 250 ms period after tone onset, normalized to the mean activity during the 250 ms period directly before the tone.
DOI: https://doi.org/10.7554/eLife.43281.010

The following source data and figure supplements are available for figure 3:

**Source data 1.** P values of statistical comparisons of the median spike responses to the tones.
DOI: https://doi.org/10.7554/eLife.43281.013

**Source data 2.** P values of statistical comparisons of the median local field potential responses to the tones.
DOI: https://doi.org/10.7554/eLife.43281.014

**Figure supplement 1.** Strongest neuronal responses to 1 kHz tones that signaled the no-go response.
DOI: https://doi.org/10.7554/eLife.43281.011

**Figure supplement 2.** Comparison of neuronal responses to tones of the same frequency in the passive condition.
DOI: https://doi.org/10.7554/eLife.43281.012

curves; for detailed p values, see *Figure 3—source data 1*). The corresponding differences in the normalized spike responses in the passive condition were smaller (*Figure 3—figure supplement 2A*). These results corroborate our earlier conclusion that the differences in the normalized spike responses during task performance cannot be fully explained by task-independent stimulus-specific adaptation or forward suppression of the spike responses. These findings show that the spike response to a given tone in auditory cortex was strongest when the tone, irrespective of whether it was presented as S1 or as S2, signaled that the no-go response was required in a trial to obtain a reward. In addition, our finding that the spike responses to S1 and S2 in the same trial differed during task performance (i.e., S1-no-go vs. S2-nil; *Figure 3A*) but not in the passive condition (*Figure 3—figure supplement 2A*) indicates that the spike response to a given tone can change rapidly, on a time scale of one second, depending on the required motor response signaled by the tone.

## Representation of sensorimotor associations in auditory cortex is related to behavioral performance

So far, we have analyzed spike responses only from trials where the monkeys performed the tasks correctly and shown that neurons in auditory cortex represent sensorimotor associations. This restriction, however, does not allow us to examine whether such representations are related to an animal's performance. Addressing this issue might provide insights into the importance of such representations for behavior. We therefore analyzed spike responses from trials where the monkeys performed the tasks incorrectly. We focused on *false-alarm trials* where the monkeys made the go response instead of the required no-go response. We computed the normalized spike responses in false-alarm trials and compared them with the responses in correct trials. These analyses were limited to the multiunits with significant spike ratios during tone presentation in correct trials (e.g., green dots in *Figure 2Ci and 2Di*) and whose responses were recorded during at least five false-alarm trials.

For S1, we compared the normalized spike responses to S1-no-go in false-alarm trials with the responses to S1-no-go in correct trials, by computing for each multiunit a ratio between the two responses. For S2, we compared the normalized spike responses to S2-no-go in false-alarm trials with the responses to S2-no-go in correct trials as well as with the responses to S2-go in correct trials, also by computing the corresponding response ratios for each multiunit. For both S1 and S2, ratios obtained for the 3 kHz and the 1 kHz tones were pooled to increase statistical power because the numbers of multiunits available for individual tones were small. This was due to the fact that the probability of false alarms was relatively low (varying across sessions from 0 to 0.35 with a median of 0.16 for the sequences with S1-no-go and from 0 to 0.33 with a median of 0.05 for the sequences with S2-no-go).

For both S1 and S2, the normalized spike responses differed between false-alarm trials and correct trials. *Figure 4A* shows the distribution of the ratios across all 168 comparisons for S1. The responses to S1-no-go in false-alarm trials were smaller than the responses to S1-no-go in correct trials, reflected in the fact that the median of the corresponding response ratios was 0.92, which was significantly <1 (p<0.05, Wilcoxon signed rank test). Similar results were obtained for the 51 comparisons for S2 such that the responses to S2-no-go in false-alarm trials were smaller than the responses to S2-no-go in correct trials (*Figure 4C*; black curve; median of 0.71; p<0.05, Wilcoxon signed rank test). In contrast, the responses to S2-no-go in false-alarm trials were not significantly different from the responses to S2-go in correct trials (gray curve; median of 0.95; p>0.05, Wilcoxon signed rank test). These analyses show that spike responses to tones that signaled the no-go response in a trial were related to the monkeys' behavior.

## Representation of sensorimotor associations in the local field potentials in auditory cortex

Analysis of the local field potentials (LFPs), which were recorded simultaneously with the spiking activity, revealed that sensorimotor associations are represented also in the activity of neuronal populations in early auditory cortex. *Figure 5A and 5B* show the LFPs at an example site while the monkey correctly performed Task 1 and Task 2. For the 3 kHz tone, the potential evoked by S1-uncertain differed from that evoked by S1-no-go (compare orange traces in *Figure 5Ai and 5Bi*). Also, the potential evoked by S2-go differed from that evoked by S2-no-go (compare orange traces in *Figure 5Aii and 5Biv*). The differences were most pronounced during the second negative

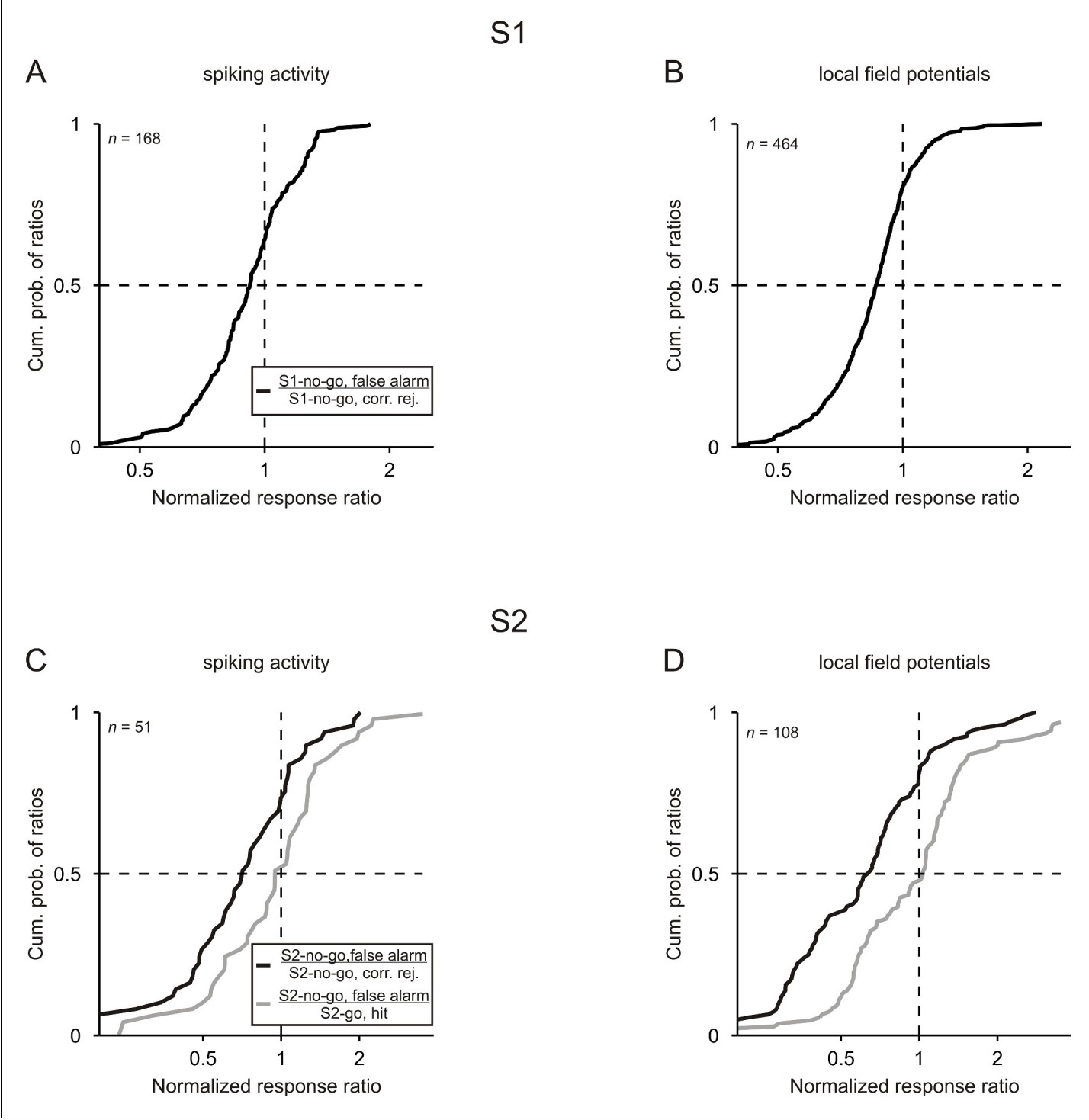

**Figure 4.** Neuronal responses are related to the behavioral performance of monkeys. Cumulative distributions of the ratios computed between the normalized responses to a given tone in false-alarm trials and in correct trials. For both S1 and S2, ratios were computed between the no-go tone in false-alarm trials and in correct trials (black curves). For S2, ratios were also computed between the no-go tone in false-alarm trials and the go tone in correct trials (gray curves). Results obtained from the 3 kHz and 1 kHz tone were pooled.

DOI: https://doi.org/10.7554/eLife.43281.015

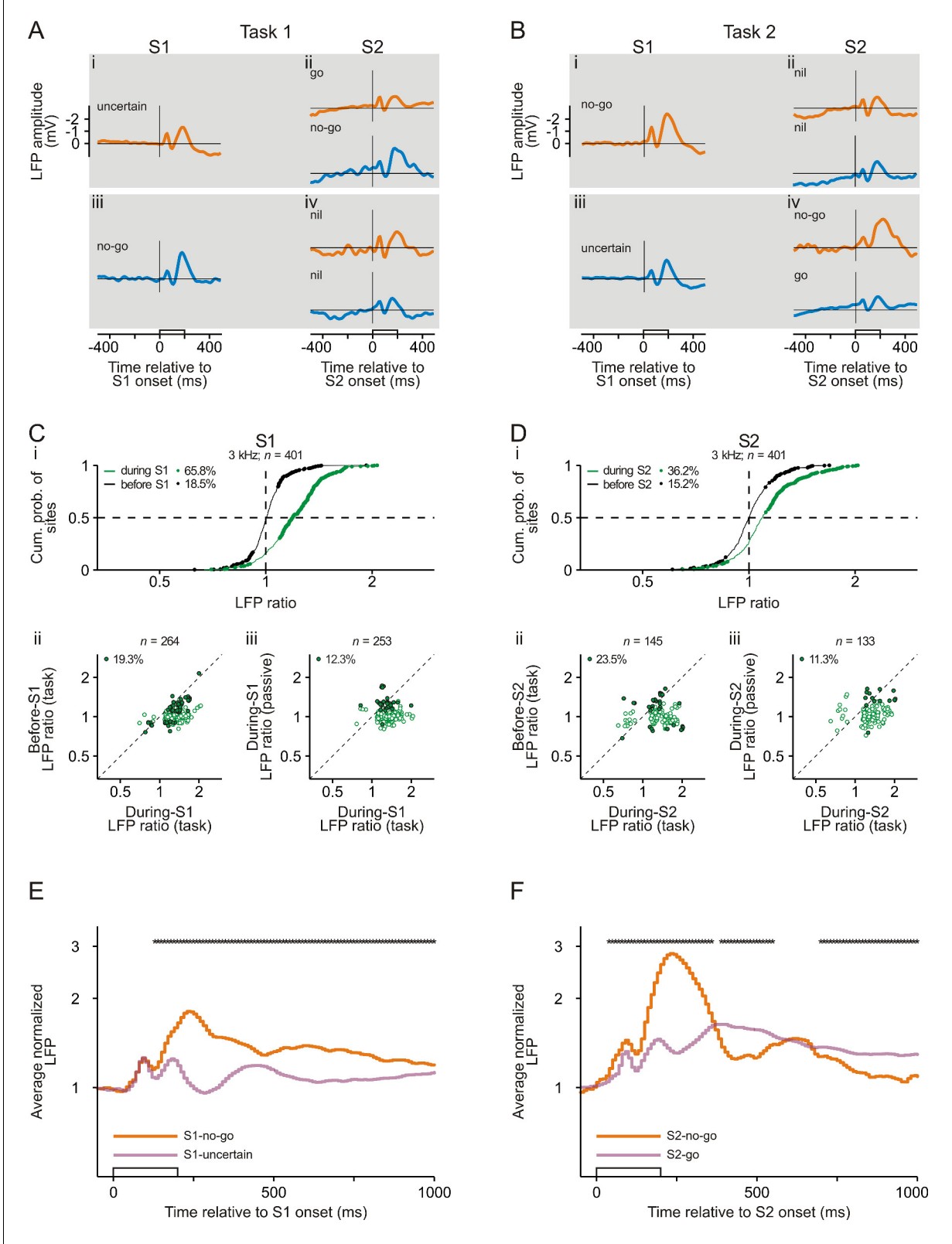

**Figure 5.** Representation of sensorimotor associations in the local field potentials in early auditory cortex. (**A, B**) The averaged local field potentials (LFPs) for an example site in auditory cortex of monkey L while she correctly performed Task 1 (**A**) and Task 2 (**B**). The horizontal lines represent an amplitude of 0 mV. Note that the ordinates in panels A and B have linear scaling. The organization of the figure and other conventions are equivalent to those of *Figure 2*.

*Figure 5 continued on next page*

*Figure 5 continued*

DOI: https://doi.org/10.7554/eLife.43281.016

The following figure supplement is available for figure 5:

**Figure supplement 1.** Local field potentials related to bar release in early auditory cortex.

DOI: https://doi.org/10.7554/eLife.43281.017

deflection of the evoked potential. To quantify these differences during the full duration of the evoked potential, we calculated the root-mean-square value of the LFP (RMS-LFP) from 0 to 500 ms after tone onset. Analogous to the analysis of the spiking activity, we then calculated the ratio of the RMS-LFP during S1-no-go to that during S1-uncertain (during-*S1 LFP ratio*) and the ratio of the RMS-LFP during S2-no-go to that during S2-go (during-*S2 LFP ratio*). For this site, both during-S1 and during-S2 LFP ratios were significantly >1 (p<0.05, permutation test). This was not the case for the LFP ratios computed from the corresponding 500 ms periods directly before S1 (*before-S1 LFP ratio*) or directly before S2 (*before-S2 LFP ratio*).

For the 3 kHz tone, we obtained significant during-S1 LFP ratios at 264 of 401 studied sites (green dots in *Figure 5Ci*) and significant during-S2 LFP ratios at 145 sites (green dots in *Figure 5Di*), with the vast majorities of these ratios being >1. By analyzing before-S1 and before-S2 LFP ratios (black curves and dots in *Figure 5Ci and 5Di*), we estimated that ~20% of the significant during-S1 and during-S2 LFP ratios could have been caused by the difference in the visual cue (filled dots in *Figure 5Cii and 5Dii*). By analyzing during-S1 and during-S2 LFP ratios in the passive condition, we estimated that ~10% of the significant during-S1 and during-S2 LFP ratios obtained during task performance could have been caused by the differences in the acoustic context (filled dots in *Figure 5Ciii and 5Diii*). Furthermore, during task performance, the proportion of sites having significant during-S1 LFP ratios >1 or during-S2 LFP ratios >1 did not consistently vary with the difference in the tone probability in the direction expected from task-independent stimulus-specific adaptation of the neuronal responses (*Figure 2—figure supplement 3B*), providing additional support that these significant LFP ratios were not solely due to the difference in the acoustic context between the two tasks. Thus, about 70% of the significant during-S1 LFP ratios (45.0% of all 401 sites) and 70% of the significant during-S2 LFP ratios (23.6% of all 401 sites) were considered to be due to differences in the sensorimotor associations.

We also computed the average normalized LFPs for the subpopulations of sites with significant during-S1 or during-S2 LFP ratios (*Figure 5E and 5F*; for details of the calculation of the average normalized LFPs, see Materials and methods). In addition, for the subpopulation of sites that exhibited either significant during-S1 LFP ratios or significant during-S2 LFP ratios or both, we compared the normalized LFP to the same tone when it was presented as S1-no-go, S1-uncertain, S2-no-go, S2-go or S2-nil. We found that the LFPs represented sensorimotor associations in a way similar to the spike rates: (1) the average normalized LFPs for S1-no-go and S1-uncertain and for S2-no-go and S2-go started to differ shortly after the beginning of the early part of the response and continued throughout the late part of the response (*Figure 5E and 5F*); (2) the normalized LFP to a given tone was strongest when the tone, irrespective of whether it was presented as S1 or as S2, signaled that the no-go response was required in a trial (*Figure 3B*; p<0.05/4, each corrected for multiple pairwise comparisons, Wilcoxon signed rank test; for detailed p values, see *Figure 3—source data 2*); (3) the normalized LFP to a given tone could change rapidly within a trial on a time scale of about one second depending on its sensorimotor associations, with the LFP being stronger when it was S1-no-go than when it was S2-nil (p<0.05/4; *Figure 3B*); (4) the normalized LFP to the tone that signaled the no-go response depended on the monkeys' behavior (*Figure 4B and 4D*). Compared to the average normalized spike rates *Figure 2E and 2F*, the average normalized LFPs for S1-no-go and S1-uncertain and for S2-no-go and S2-go started to differ later, the differences peaked later (~250 ms) after tone onset and ended later, but differently for S1 and S2 (*Figure 5E and 5F*). We refrained from quantitatively comparing the temporal relationships between the spike activity and LFPs because LFPs reflect the activity of large neuronal populations and because a longer time interval was used to identify the sites with significant LFP ratios. Because of the use of a 500 ms time interval, the identification of significant during-S2 LFP ratios could be compromised by neuronal activity related to

the bar release after S2-go (purple trace in *Figure 5F*, ~400 ms after the onset of S2-go; *Figure 5—figure supplement 1A*).

## Discussion

We have shown here that early auditory cortex of nonhuman primates represents sensorimotor associations. Changes of sensorimotor associations were paralleled by changes of spike responses and LFP responses to tones, and the neuronal responses to a given tone were largest when the tone signaled that the no-go response was required in a trial to obtain a reward. These changes of the neuronal responses were observed shortly after the beginning of the tone-evoked responses.

### Separation of sensorimotor associations from other task components

We investigated whether auditory cortex represents sensorimotor associations by comparing neuronal activity in two tasks which differed with respect to the motor response that was signaled by a given tone. Because the tasks differed also in other aspects that could affect neuronal activity in auditory cortex, we used several approaches to estimate the potential contributions of these confounds to the differences in neuronal activity between the tasks and to show that neuronal activity in auditory cortex is related to sensorimotor associations. These confounds were the visual cue, the acoustic context, the motor response required and executed after the tone, and the level of attention. Because none of these potential confounds could fully account for the differences in the neuronal activity between the two tasks, our study demonstrates that auditory cortex represents sensorimotor associations. Previous studies that investigated this issue did not control for or estimate potential contributions of various confounds such as the cue (*Vaadia et al., 1982*), the acoustic context (*Jaramillo et al., 2014*) or the valence of sounds (*David et al., 2012*). The latter confound was not present in our study due to the symmetrical reward, that is the monkeys received the same reward for correct go responses and correct no-go responses.

### Characteristics of the representation of sensorimotor associations in auditory cortex

Cortical representations of sensorimotor associations are commonly studied by comparing the neuronal responses to a given stimulus when it signals different motor responses in different tasks (e.g., *Asaad et al., 1998*; *David et al., 2012*; *Jaramillo et al., 2014*; *Sakagami and Tsutsui, 1999*; *Sakagami et al., 2001*; *Vaadia et al., 1982*). Here, we modified this approach and used sequences of two stimuli instead of a single stimulus. This was done to introduce more than two sensorimotor associations for the same stimulus in the same task and to introduce different sensorimotor associations for a given stimulus depending on its position in a sequence (see *Figure 1C and 1D*). We found that the neuronal responses in auditory cortex differed depending on whether a tone could resolve the uncertainty regarding the required motor response and thus could signal the motor response (S1-no-go) or not (S1-uncertain); they also differed depending on whether a tone signaled the go response (S2-go) or the no-go response (S2-no-go) to obtain a reward or signaled the reward value of a given motor response. In our study, the response differences to the S1s and to the S2s may not be fully independent, because S1-no-go not only resolved the uncertainty regarding the required motor response but also signaled that the no-go response was required in a trial and because the neuronal responses were always largest to the no-go tone, whether S1 or S2 (*Figure 3*, *Figure 3—figure supplement 1*). The observation that the neuronal responses to the no-go tone were largest shows that neurons in auditory cortex respond differently to a given tone when it is repeated in a trial but the sensorimotor association differs, for example to S1-no-go and to S2-nil in the same sequence (*Figure 1C*, bottom; *Figure 1D*, top). Therefore, the responses of neurons in auditory cortex can rapidly change with the required motor response signaled by the tone, that is with the behavioral meaning of the tone.

The existence of a relationship between neuronal responses and behavioral meaning is further supported by our observation that neuronal responses in false-alarm trials were similar to responses in correct go trials (*Figure 4*). Therefore, the neuronal responses in auditory cortex did not reflect what the monkeys should have done, but rather reflected what the monkeys actually did. Our study thus provides further support that neuronal activity in auditory cortex can be related to an animal's

behavior (e.g., *Bizley et al., 2013*; *Niwa et al., 2012*; *Selezneva et al., 2006*; *Tsunada et al., 2016*).

We demonstrated that sensorimotor associations were represented in auditory cortex in specific time periods, that is within the 250 ms period after tone onset for the spiking activity and within the 500 ms period for the LFPs. It is possible that sensorimotor associations were also present in auditory cortex during other periods in a trial, that is before S1, during the delay before S2, and after S2. For both S1 and S2, sensorimotor associations may already be present in auditory cortex before tone presentation, because the monkeys needed to know the task rules, that is which motor response to perform after which tone sequence, to select the appropriate motor response. After S2, sensorimotor associations were needed at least until the moment when they were converted into actual motor signals. During each of these three periods, we observed differences in the neuronal activity between the two tasks (e.g., *Figures 2* and *5*; also see *Huang et al., 2016a*), but our experimental design was not optimized to control for potentials confounds and to clarify whether these differences in the neuronal activity were related to sensorimotor associations.

## How does the representation of sensorimotor associations in auditory cortex emerge?

For both S1 and S2, changes of sensorimotor associations were paralleled by changes of the tone-evoked neuronal responses shortly after the beginning of the responses (*Figure 2E and 2F*). This suggests that the auditory system contributes to the association of a tone with a motor response. To accomplish this, the activity in the auditory system must change in a task-specific way based on the knowledge about the required sensorimotor associations. Such knowledge could be acquired and stored in long-term memory during the training stages and retrieved for use in working memory during task performance. The task-specific changes of the neuronal activity in auditory cortex may result in an enhanced contrast of the neuronal responses to the tones having different meanings in the two tasks, for example by increasing the neuronal responses to a no-go tone. Such changes may arise from task- and meaning-specific changes of stimulus-specific adaptation, forward suppression, or receptive field properties of neurons in auditory cortex or in earlier stages of the auditory system (*Atiani et al., 2009*; *Groh et al., 2001*; *Gruters et al., 2018*; *Metzger et al., 2006*). These changes may be mediated by sub-cortical neuromodulator systems, for example cholinergic and dopaminergic nuclei (*Ayala and Malmierca, 2015*; *Huang et al., 2016b*), or by higher-order associative systems, for example prefrontal cortex (*Gao et al., 2017*; *Winkowski et al., 2013*; *Winkowski et al., 2018*).

## Role of auditory cortex in the selection of motor responses

The results of our study suggest that early auditory cortex may contribute to the selection of appropriate motor responses to auditory stimuli. Therefore, early auditory cortex appears to be involved in a function that traditionally has been associated with parietal, frontal, and prefrontal cortices (*Asaad et al., 1998*; *Freedman and Ibos, 2018*; *Komatsu, 1982*; *Sakagami and Tsutsui, 1999*; *Sakagami et al., 2001*; *Watanabe, 1986*; *Yamatani et al., 1990*). Our results thus point to a possible communication between auditory cortex and association cortices (*Sheikhattar et al., 2018*). Such communication may contribute to the selection of appropriate motor responses to auditory stimuli. It is conceivable that neuronal representations of sensorimotor associations in auditory cortex exist not only for the tasks used here but also for other tasks, such as reacting to the honking of a car in the middle of a crosswalk. It is also conceivable that sensorimotor associations are represented in other sensory cortices for other stimuli, for example in visual cortex for visual stimuli or in somatosensory cortex for tactile stimuli. It has been shown that neuronal activity in visual and somatosensory cortices can be affected by reinforcement (*Baruni et al., 2015*; *McNiel et al., 2016*; *Pantoja et al., 2007*; *Shuler and Bear, 2006*) which plays crucial roles in shaping sensorimotor associations.

We observed that the responses of neurons in auditory cortex to tones were enhanced when the tones signaled that the no-go response was required in a trial or, phrased differently, that the go response was not required. This observation might point to a role of auditory cortex in inhibitory control, a process that permits individuals to inhibit their natural or dominant motor response in order to select a response that is consistent with completing their current goals (*Chambers et al.,*

*2009*; *Diamond, 2013*). In our study, inhibitory control may have been required for the go response which the monkeys had to execute more frequently so that was the dominant motor response. Our results therefore extend the knowledge about the neuronal basis of inhibitory control, which has been obtained mainly by conducting studies in frontal and prefrontal regions (e.g., *Hodgson et al., 2007*; *Konishi et al., 1998*; *Sakagami et al., 2001*).

In conclusion, our findings add to other non-auditory functions of auditory cortex, such as representing non-auditory stimuli, motor behavior, and motivational aspects of tasks (*Bizley and King, 2008*; *Brosch et al., 2005*; *Brosch et al., 2015*; *Eliades and Wang, 2008*; *Niwa et al., 2012*) and temporarily storing task-relevant information (*Huang et al., 2016a*). These functions may be crucial for auditory cortex to integrate diverse task-relevant information to adapt to task demands and thus to serve as a semantic processor (*Scheich et al., 2011*) or a problem solver (*Weinberger, 2011*), ultimately contributing to the selection and preparation of appropriate motor responses to achieve a given goal in different situations.

## Materials and methods

Materials and methods were largely identical to those described earlier (*Huang et al., 2016a*). The experiments were approved by the authority for animal care and ethics of the federal state of Saxony-Anhalt (No. 28-42502-2-1129IfN), and conformed to the rules for animal experimentation of the European Community Council Directive (86/609/EEC).

### Behavioral paradigms

Two monkeys (C and L; Macaca fascicularis) performed two tasks on a set of four two-tone sequences within the same experimental sessions. Each trial started with the illumination of a light-emitting diode (LED), which signaled that the monkey had to grasp a bar within 4 s and to hold it subsequently (*Figure 1A*). One to two seconds after the monkey had grasped the bar, two tones (S1 and S2) were presented bilaterally through loudspeakers (Karat 720), located to the monkey's left and right. The sound pressure level of each tone was 67 dB SPL and the duration was 200 ms. The stimulus-onset interval was 1000 ms, resulting in a delay of 800 ms between the offset of S1 and the onset of S2. The frequency of each tone was either 3 kHz or 1 kHz. When the go response was required in a trial, monkey L had to release the bar within 40–1160 ms after the offset of S2 and monkey C within 40–1760 ms. When the no-go response was required, the monkeys had to continue holding the bar until the end of these time windows. Upon correct go responses and correct no-go responses, the monkeys were rewarded with a small amount of water (0.2–0.3 ml) and the LED was turned off. The next trial started if the monkeys had not touched the bar for at least 5 s. The monkeys performed the tasks using their left hand.

For Task 1, the monkeys had to make the go response when both S1 and S2 had a frequency of 3 kHz (3–3) and the no-go response for the other three sequences (3–1, 1–3, and 1–1; *Figure 1B*). For Task 2, the monkeys had to make the go response when both S1 and S2 had a frequency of 1 kHz (1–1) and the no-go response for the other three sequences (1–3, 3–1, and 3–3). In each experimental session, the monkeys performed the two tasks in separate, alternating blocks of ~140 trials (2–8 blocks, with a median of 4). The order of the blocks was randomized within each session, except that the starting task was counterbalanced across sessions. In each block, trials requiring the go response or the no-go response were presented in pseudorandom order, with the go response required in ~60% of the trials and the no-go response required in ~40% of the trials. The actual percentages of go trials and no-go trials varied across blocks. A green LED to the monkey's right signaled that Task 1 had to be performed, and a red LED to the monkey's left signaled that Task 2 had to be performed.

The two monkeys were initially trained to grasp, hold, and then release the bar to obtain a reward. Using four major steps, the monkeys were then trained to perform Task 1 and Task 2 in the same session. In the first step, the monkeys learned to make the required motor response to each of the four sequences in each of the two tasks on separate days, that is in separate training sessions. In the second step, the monkeys were trained to perform Task 1 on the 3–3 and 3–1 sequences and to perform Task 2 on the 1–1 and 1–3 sequences. The monkeys were trained on the two tasks in separate blocks on the same days for most of the time or on alternating days otherwise. For each task, and at the beginning of the training session, the sequences requiring the go response and the no-go

response were presented in separate, alternating blocks of ~50 trials. Subsequently, the block size was gradually decreased until the sequences were ultimately presented in pseudorandom order. After the monkeys had performed the two tasks correctly in ≥75% of the trials for each sequence on five consecutive days, they were trained, in the third step, to perform Task 1 on the 3–3, 3–1, and 1–3 sequences and to perform Task 2 on the 1–1, 1–3, and 3–1 sequences on the same day. In the final step, the monkeys were trained to perform both Task 1 and Task 2 on all four sequences (3–3, 3–1, 1–3, and 1–1). The training strategies for the third and the fourth step were similar to those for the second step. After ~300 training sessions for each monkey, they performed both tasks correctly in ≥65% of the trials for each of the four sequences during ten consecutive sessions.

### Passive paradigms

In the passive condition, the same set of tone sequences was presented in blocks when the monkeys did not perform the tasks. In such blocks, the LEDs were not illuminated, the touch bar was not functional, and no water was delivered. In about half of these blocks, the 3–3 sequence occurred in ~60% of the trials and the other three sequences in ~40% of the trials, as in Task 1. In the other blocks, the 1–1 sequence occurred in ~60% of the trials and the other three sequences in ~40% of the trials, as in Task 2. These passive conditions were run after the monkeys had performed the tasks or on days when they did not perform the tasks.

In an additional passive block, pure tones with 40 different frequencies were presented at ~60 dB SPL to assess the best frequency of each multiunit. Frequencies were equidistantly spaced on a logarithmic scale over a range of 8 octaves (0.0625–16 kHz). Tones had a duration of 100 ms and were presented in a pseudorandom order with a stimulus-onset interval of 500 ms until each tone had been presented ten times. This passive block was run before the monkeys performed the tasks or before the passive condition on days when the monkeys did not perform the tasks.

### Electrophysiological recordings and data analyses

Action potentials (0.5–5 kHz) and local field potentials (1–140 Hz) were simultaneously recorded from up to seven sites in the core fields of the right auditory cortex (mainly from the primary auditory field A1), using a seven-channel manipulator (2–2.5 MΩ; for details, see *Brosch et al., 2005*). Action potentials and local field potentials were also recorded from the ventrolateral part of the left prefrontal cortex in monkey C, but these data are not reported here. Action potentials of a few neurons (multiunit) recorded by a given electrode were discriminated, and the time stamp and the waveform of each action potential was stored, using the built-in spike detection tools of the data acquisition systems (threshold crossings and spike duration). Single-unit activity was extracted off-line from the multiunit recordings by means of principal component analysis. The position of A1 was estimated using the spatial distribution of the best frequencies across the recording sites (*Brosch et al., 2005*; *Kaas and Hackett, 2000*).

Data analysis was performed off-line using MATLAB (MathWorks, Natick, MA; RRID: SCR_001622). For each site, statistical analyses were based on permutation tests of the spiking activity during the 250 ms intervals before or after tone onset and of the local field potentials during the 500 ms intervals before or after tone onset. During task performance and in the corresponding passive condition, only those sites that were responsive to the tone were used for analyses. A site was considered responsive if the neuronal activity before and after tone onset differed significantly ($p < 0.05$) in at least one of the following conditions: S1-no-go, S1-uncertain, S2-no-go, or S2-go or when it was presented in the corresponding four positions in the passive condition. This preselection of sites was conducted separately for the task and passive conditions, separately for spiking activity and local field potentials, and separately for 3 kHz tones and 1 kHz tones. To calculate the average normalized LFPs (*Figure 5E and 5F*), we first computed, for each site, the root-mean-square values (RMS) of the LFP in all consecutive 10 ms bins from 500 ms directly before tone onset to 800 ms after tone offset, and then normalized the RMS value in each 10 ms bin after tone onset to the mean RMS value during the 500ms period before tone onset. Finally, these normalized LFPs were geometrically averaged across sites. For each multiunit, the best frequency was determined from the responses to the 40 pure tones as described in *Brosch et al. (1998)* and *Brosch et al. (1999)*.

## Acknowledgements

We thank Cornelia Bucks for technical assistance with the experiments, Dr. Reinhard König for participating in the design of the experiments, and Drs. Jonathan Fritz, Reinhard König, and Liang Li for their valuable comments on the manuscript. This research was supported by a CBBS Neuronetwork Project, a LIN Special Project, and projects funded by the Deutsche Forschungsgemeinschaft (He 1721/10-1, He 1721/10-2, and SFB TR31, A4, and A6). The publication of this article was funded by the Open Access Fund of the Leibniz Association.

## Additional information

### Funding

| Funder | Grant reference number | Author |
| --- | --- | --- |
| European Regional Development Fund | CBBS neuronetwork | Ying Huang |
| Deutsche Forschungsgemeinschaft | He 1721/10-1 | Peter Heil<br>Michael Brosch |
| LIN Special Project | | Peter Heil<br>Michael Brosch |
| Deutsche Forschungsgemeinschaft | He 1721/10-2 | Peter Heil<br>Michael Brosch |
| The Open Access Fund of the Leibniz Association | | Ying Huang<br>Peter Heil<br>Michael Brosch |

The funders had no role in study design, data collection and interpretation, or the decision to submit the work for publication.

### Author contributions

Ying Huang, Conceptualization, Resources, Data curation, Software, Formal analysis, Supervision, Funding acquisition, Validation, Investigation, Visualization, Methodology, Writing—original draft, Project administration, Writing—review and editing; Peter Heil, Michael Brosch, Conceptualization, Resources, Supervision, Funding acquisition, Validation, Visualization, Writing—original draft, Project administration, Writing—review and editing

### Author ORCIDs

Ying Huang ORCID http://orcid.org/0000-0002-6471-8009
Peter Heil ORCID http://orcid.org/0000-0001-7861-5927

### Ethics

Animal experimentation: The experiments in this study were approved by the authority for animal care and ethics of the federal state of Saxony-Anhalt (No. 28-42502-2-1129IfN), and conformed to the rules for animal experimentation of the European Community Council Directive (86/609/EEC).

### Decision letter and Author response

Decision letter https://doi.org/10.7554/eLife.43281.020
Author response https://doi.org/10.7554/eLife.43281.021

## Additional files

### Supplementary files

• Transparent reporting form
DOI: https://doi.org/10.7554/eLife.43281.018

## Data availability

All data generated or analysed during this study are included in the manuscript and supporting files.

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
