## [Decision Letter]

Thank you for submitting your article "Associations between sounds and actions in early auditory cortex of nonhuman primates" for consideration by *eLife*. Your article has been reviewed by three peer reviewers, and the evaluation has been overseen by a Reviewing Editor and Timothy Behrens as the Senior Editor. The following individuals involved in review of your submission have agreed to reveal their identity: Stephen V. David (Reviewer #1); Christopher I Petkov (Reviewer #2).

The reviewers have discussed the reviews with one another and the Reviewing Editor has drafted this decision to help you prepare a revised submission.

This study by Huang and colleagues is timely and addresses important questions.

Summary:

This paper set out to establish motor related signals in auditory cortex. The authors trained two monkeys to switch between go and no-go responses within the same testing session, with the response depending on a sequence of two tones. They recorded spiking activity and local field potentials in both auditory cortex and ventro-lateral prefrontal cortex. Other studies from rodent and ferret auditory cortex have suggested that there might be motor related signals, but the authors are correct that those conclusions were tentative because a number of alternative explanations were not ruled out. This study focuses on these alternative explanations and attempts to address them.

Overall, this manuscript is well written, interesting, timely and will help establish the existence of non-auditory signals in auditory cortex.

Essential revisions:

1) Several reviewers felt that the authors had not firmly established the presence of motor related signals in auditory cortex, or that alternative explanations had not been fully eliminated. They suggested an additional analysis and a change in the way some of the claims were discussed.

For example, when a tone shifts from go to no-go, the required motor response changes, but the reward associated with a given motor response also changes. That is, in a context where S1 means "punish for go", an enhanced AC response to S1 could signal that a go response to that sound is associated with a negative reward. While overall reward is the same for Task 1 and Task 2, the reward associated with the *same* motor response to the *same* stimulus is changing. Regarding questions of reward, consider recent work from the Salzman group (Baruni et al., 2015) in visual cortex where apparent selective attention effects could in fact be explained by associated reward value. One might also consider efforts to dissociate motor and category representations in visual prefrontal work (e.g., Freedman and Ibos, 2018). It appears that reward value changes in the current manipulation. The authors should carefully consider possible confounds in their task and highlight limitations to the interpretation of their results.

The central claim of the manuscript could be strengthened by examining error trial neural activity. For example, neural data on hit and false-alarm trials should be comparable. Stated another way, in error trials, is the activity closer to what the animal should have done, or what it did do? Does the activity in error trials encode the action, when the action should have been withheld? This would provide further evidence supporting encoding of motor signals. In addition to exploring this question here, the authors should discuss the relevant studies, namely Bizley et al. (2013) and Tsunada et al. (2016).

2) Moreover, it is not clear that the controls for visual cue/task condition are complete. This concern has to do with pre-stimulus effects of task condition. The authors report a small but possibly significant change in spike rate prior to S1 onset (Figure 2Cii). This effect appears to be a tonic change in firing that reflects the rules of the current block (see Rodgers and DeWeese, 2014). It seems like such a tonic change should continue through the sound evoked period. Thus, changes in the S1 response should be measured after correcting for the change in pre-stimulus spike rate. It doesn't seem that this will critically affect the main result, but it is important that the authors isolate the change in sound-evoked response from tonic changes in spike rate.

The data can also be interpreted as a change in attention, as opposed to sensorimotor interactions. That is, activity may decrease on trials when attention to the second stimulus is required to determine task outcome, relative to those trials in which attention is not required because the outcome is known. Potentially, both are interesting but the interpretation of the data and ultimately, AC's function would be different.

The interpretation that stronger no-go responses in auditory cortex (which interestingly reverse to stronger go responses, as expected, later when the response needs to be made; Figure 2Fii) are related to (Abstract) "inhibiting inappropriate responses during goal-directed behaviour" is unclear. The Discussion does not provide more clarity on what the authors mean, so I do not see what the authors mean by "inhibiting" or "inappropriate" responses. This is crucial for the conclusion, and it might be useful to relate to or distinguish from Eliades and Wang (2008) where they see both suppression and enhancement in auditory cortex during vocalization in marmosets. Both vocalization and pressing a lever are self-generated movements but the movements are vastly different, of course. Again, analysis of the error trials might clarify the conclusion.

3) The vlPFC data and their interpretation of it vis-a-vis AC activity should be removed from the manuscript. Data from one animal strongly limits the conclusions that can be drawn.

---

## [Author Response]

Essential revisions:1) Several reviewers felt that the authors had not firmly established the presence of motor related signals in auditory cortex, or that alternative explanations had not been fully eliminated. They suggested an additional analysis and a change in the way some of the claims were discussed.

We would like to thank the reviewers for pointing out the alternative explanations of our results. We addressed the reviewers’ concerns and performed additional analyses as suggested and as explained in detail below.

For example, when a tone shifts from go to no-go, the required motor response changes, but the reward associated with a given motor response also changes. That is, in a context where S1 means "punish for go", an enhanced AC response to S1 could signal that a go response to that sound is associated with a negative reward. While overall reward is the same for Task 1 and Task 2, the reward associated with the same motor response to the same stimulus is changing. Regarding questions of reward, consider recent work from the Salzman group (Baruni et al., 2015) in visual cortex where apparent selective attention effects could in fact be explained by associated reward value. One might also consider efforts to dissociate motor and category representations in visual prefrontal work (e.g., Freedman and Ibos, 2018). It appears that reward value changes in the current manipulation. The authors should carefully consider possible confounds in their task and highlight limitations to the interpretation of their results.

We agree with the reviewers that the behavioral meaning of a tone can be formulated in two different, albeit equivalent, ways. For example, S2-go means reward for the go-response and equivalently no reward for the no-go response. Similarly, S2-no-go means reward for the no-go-response and equivalently no reward for the go response. S2-go and S2-no-go therefore differed with respect to the motor response required to obtain a reward (e.g., go: reward vs. no-go: reward) and equivalently with respect to the reward value of a given motor response (e.g., go: reward vs. go: no reward). Our experimental design cannot distinguish between these two equivalent formulations of a stimulus-action-outcome association. This is now explained in the third paragraph of the subsection “Experimental rationale and overview”. In addition, as the reviewers suggested, we cited the study of Baruni et al. (2015) in the first paragraph of the subsection “Role of auditory cortex in the selection of motor responses”, as well as the study of Freedman and Ibos (2018).

The central claim of the manuscript could be strengthened by examining error trial neural activity. For example, neural data on hit and false-alarm trials should be comparable. Stated another way, in error trials, is the activity closer to what the animal should have done, or what it did do? Does the activity in error trials encode the action, when the action should have been withheld? This would provide further evidence supporting encoding of motor signals. In addition to exploring this question here, the authors should discuss the relevant studies, namely Bizley et al. (2013) and Tsunada et al. (2016).

As suggested, we included the results of additional analyses of both spiking activity and LFPs in error trials. We found that neuronal responses to tones in error trials reflected what the monkeys actually did rather than what they should have done, supporting our central suggestion that there are motor-related signals in early auditory cortex. To show these results, we added a new figure (Figure 4) and a new subsection (“Representations of sensorimotor associations in auditory cortex is related to behavioral performance”), as well as a new sentence in the last paragraph of the subsection “Representation of sensorimotor associations in the local field potentials in auditory cortex”. In addition, we added a new paragraph in the Discussion (subsection “Characteristics of the representation of sensorimotor associations in auditory cortex”, second paragraph) to discuss the relevant studies including Bizley et al. (2013) and Tsunada et al. (2016).

2) Moreover, it is not clear that the controls for visual cue/task condition are complete. This concern has to do with pre-stimulus effects of task condition. The authors report a small but possibly significant change in spike rate prior to S1 onset (Figure 2Cii). This effect appears to be a tonic change in firing that reflects the rules of the current block (see Rodgers and DeWeese, 2014). It seems like such a tonic change should continue through the sound evoked period. Thus, changes in the S1 response should be measured after correcting for the change in pre-stimulus spike rate. It doesn't seem that this will critically affect the main result, but it is important that the authors isolate the change in sound-evoked response from tonic changes in spike rate.

We apologize that we were obviously not clear enough in our original manuscript regarding the controls we performed for the neuronal activity related to visual cue/task condition. Specifically, we used two approaches to control for tonic pre-stimulus differences in neuronal activity related to visual cue/task condition. First, we estimated the percentage of units whose activity differed between the two tasks both before tone presentation and during tone presentation. The number of these units was subtracted from the number of units whose activity differed significantly during tone presentation when estimating the percentage of units that represent sensorimotor associations. Our second approach was to compare the activity during tone presentation between the two tasks after correcting for any differences in activity before tone presentation by normalizing the activity during tone presentation to the activity before tone presentation, as suggested by the reviewers. The results of this approach had been shown in Figure 2E-2F, Figure 2—figure supplement 1E-F, Figure 3, and Figure 3—figure supplement 1. We tried to make this clearer to readers by modifying the text (subsection “Experimental rationale and overview”, sixth paragraph, subsection “Representation of sensorimotor associations in auditory cortex revealed by comparing spike responses to S1”, third and last paragraphs and subsection “Spike responses were strongest to tones that signaled the no-go response”, first paragraph). In addition, we cited the study of Rodgers and DeWeese (2014) in the second paragraph of the Introduction and subsection “Experimental rationale and overview”, sixth paragraph.

The data can also be interpreted as a change in attention, as opposed to sensorimotor interactions. That is, activity may decrease on trials when attention to the second stimulus is required to determine task outcome, relative to those trials in which attention is not required because the outcome is known. Potentially, both are interesting but the interpretation of the data and ultimately, AC's function would be different.

We agree with the reviewers that the level of attention to S2 might differ depending on whether S2 was required to solve the task or not. However, the differences in the neuronal responses to S2 observed in our study cannot be fully explained by a change in attention. First, we compared neuronal responses to S2-go and S2-no-go. Both were required to solve the task and thus likely did not differ in level of attention devoted to them, but the neuronal responses to differed. Second, the level of attention devoted to both S2-go and S2-no-go might have differed from that devoted to S2-nil because S2-nil was not required to solve the task. Nevertheless, differences in the neuronal response were observed only between S2-no-go and S2-nil but not between S2-go and S2-nil (Figure 2—figure supplement 5). This suggests that the differences in the neuronal responses to S2-no-go and S2-nil were unlikely due to differences in attention, because otherwise similar differences in the neuronal response should have been observed also between S2-go and S2-nil. In addition, the difference in the neuronal response to S1 between the two tasks also cannot be explained by differences in the level of attention because both S1-no-go and S1-uncertain were required to solve the task and therefore presumably did not differ in the level of attention devoted to them. To clarify this issue, we made changes in the subsection regarding the experimental rationale (“Experimental rationale and overview”), in the Results subsection (“Representation of sensorimotor associations in auditory cortex revealed by comparing spike responses to S2”, last paragraph).

The interpretation that stronger no-go responses in auditory cortex (which interestingly reverse to stronger go responses, as expected, later when the response needs to be made; Figure 2Fii) are related to (Abstract) "inhibiting inappropriate responses during goal-directed behaviour" is unclear. The Discussion does not provide more clarity on what the authors mean, so I do not see what the authors mean by "inhibiting" or "inappropriate" responses. This is crucial for the conclusion, and it might be useful to relate to or distinguish from Eliades and Wang (2008) where they see both suppression and enhancement in auditory cortex during vocalization in marmosets. Both vocalization and pressing a lever are self-generated movements but the movements are vastly different, of course. Again, analysis of the error trials might clarify the conclusion.

We apologize for using the misleading language in this instance. The term ‘inhibiting inappropriate responses’ was not precisely used here. We meant to interpret the stronger neuronal responses to the no-go tone in such a way that auditory cortex can provide a neuronal signal to indicate that a certain motor response (in our case, the go response) needs to be inhibited in the upcoming future, thus contributing to the selection of the appropriate motor responses to sounds. The enhanced neuronal responses to the no-go tone are related to the behavioral meaning of the tone and therefore different from the neuronal activity related to the actual execution of motor behavior such as grasping a bar (e.g., Brosch et al., 2005), releasing a bar (e.g., Figure 2F in our study), and vocalizing (e.g., Eliade and Wang, 2008). To avoid confusion, we removed the term ‘inhibiting inappropriate responses’ from the manuscript. Furthermore, to contrast the neuronal activity related to the behavioral meaning of a given tone and the neuronal activity related to the actual execution of a motor response, i.e., the bar release in our study, we added two new figures to show the neuronal activity in auditory cortex related to the bar release (Figure 2—figure supplement 4 for the spiking activity and Figure 5—figure supplement 1 for LFPs). We also made changes in the sixth paragraph of the subsection “Representation of sensorimotor associations in auditory cortex revealed by comparing spike responses to S2” to describe the neuronal activity related to this bar release. In addition, we cited the study of Eliades and Wang (2008) in the fifth paragraph of the subsection “Experimental rationale and overview” and in the last paragraph of the subsection “Role of auditory cortex in the selection of motor responses”.

As suggested here and also in Essential revision 1, we performed additional analyses for error trials. For details of the analyses and the results, see our responses to Essential revision 1 starting with “The central claim of the manuscript could be strengthened by examining error trial neural activity”.

3) The vlPFC data and their interpretation of it vis-a-vis AC activity should be removed from the manuscript. Data from one animal strongly limits the conclusions that can be drawn.

As suggested, we removed the results obtained from vlPFC and the comparison between vlPFC and AC from the manuscript.